# The Microbial Community Structure in the Rhizosphere of *Theobroma cacao* L. and *Euterpe oleracea* Mart. Is Influenced by Agriculture System in the Brazilian Amazon

**DOI:** 10.3390/microorganisms12020398

**Published:** 2024-02-17

**Authors:** Rosiane do Socorro dos Reis de Sousa, Giulia Victória Silva Lima, Josinete Torres Garcias, Graziane de Oliveira Gomes, Jackeline Rossetti Mateus, Lucimar Di Paula dos Santos Madeira, Lucy Seldin, Hervé Louis Ghislain Rogez, Joana Montezano Marques

**Affiliations:** 1Instituto de Ciências Biológicas, Universidade Federal do Pará, Belém 66075-110, Pará, Brazil; rosiane.sousa@icb.ufpa.br (R.d.S.d.R.d.S.); giuliavlima@outlook.com (G.V.S.L.); josygarcias19@gmail.com (J.T.G.); graziane.gomes@icb.ufpa.br (G.d.O.G.); madeiraldps@ufpa.br (L.D.P.d.S.M.); herverogez@gmail.com (H.L.G.R.); 2Instituto de Microbiologia Paulo de Góes, Universidade Federal do Rio de Janeiro, Rio de Janeiro 21941-902, Rio de Janeiro, Brazil; jacky.rossetti@micro.ufrj.br (J.R.M.); lseldin@micro.ufrj.br (L.S.)

**Keywords:** agroforestry system, bacterial and fungal communities, PGPR

## Abstract

This study tested the hypothesis that cocoa monoculture (MS) and cocoa-açai agroforestry systems (AFS) may influence the microbial community structure and populations of plant growth-promoting bacteria (PGPR). Accordingly, the aim was to analyze the microbial community structure and PGPR populations in different agroecosystems in the Brazilian Amazon. To achieve this, the rhizosphere microbial community of cocoa and açai plants in both Amazonian seasons (dry and rainy) was analyzed using culture-dependent (PGPR screening) and -independent methods [PCR-DGGE based on *rrs*, *alp*, *nif*H gene, and intergenic region (ITS) of fungi]. Concerning PGPR screening, out of 48 isolated bacterial strains, 25% were capable of siderophore production, 29% of mineralized organic phosphate, 8% of inorganic phosphate solubilization, and 4% of indole acetic acid production. Moreover, 17% of isolates could inhibit the growth of various phytopathogenic fungi. Statistical analyses of DGGE fingerprints (*p* < 0.05) showed that bacterial and fungal community structures in the rhizosphere were influenced by the seasons, supporting the results of the physicochemical analysis of the environment. Furthermore, as hypothesized, microbial communities differed statistically when comparing the MS and AFS. These findings provide important insights into the influence of climate and cultivation systems on soil microbial communities to guide the development of sustainable agricultural practices.

## 1. Introduction

The Amazon rainforest is considered the ecosystem with the highest biodiversity on our planet. Among this great biological diversity, plants such as cocoa (*Theobroma cacao* L.) and açai (*Euterpe oleracea* Mart.) are in the spotlight due to their economic potential [1,2]. Nowadays, Brazil is the main producer of açai and the seventh largest producer of cocoa in the world. The state of Pará possesses 27.14% of the Brazilian Amazon forest and leads the national production of both crops [3,4,5].

Although cocoa and açai in the field are largely cultivated in monoculture systems (MS), alternative systems, such as the agroforestry systems (AFS), are attracting attention as cocoa-açai intercropping is expanding in the state of Pará. The AFS is a sustainable system that proposes the combined use of different crops in the same area as a forest ecosystem. This intercropping system optimizes production as different crops are planted at the same time, which reduces deforestation. In addition, AFS advantages also include social benefits for family farmers that can enhance household income-producing cocoa and açai in small farm areas [6,7,8,9].

The AFS benefits for soil health are already reported in the literature [10]. The blend of different trees can provide habitats for numerous species with different ecological functions, such as insects and birds, which can help pest and disease control, reducing, for example, the use of pesticides and chemical fertilizers. In the same way, the AFS was shown to influence the soil microbial community structure [11]. As the enhancement of carbon and energy sources were already observed in such agriculture systems, an improvement in the microbial abundance has also been reported [12,13,14]. Considering soil health, the presence of different plants in the same crop area is interesting, as plant roots can influence microorganisms’ recruitment, performing the “rhizosphere effect”. This effect can include the recruitment of bacterial strains that promote plant growth, the so-called plant growth-promoting rhizobacteria (PGPR) [15,16,17].

The recruitment of PGPR to the soil rhizosphere was demonstrated with different plant exudation [18,19]. A few studies highlighted cocoa and açai plants individually as natural hosts for several associative rhizobacteria. Among these studies, the rhizobacteria associated with both plants showed the potential to promote plant growth and/or to control plant pathogens [20,21,22,23]. Exploring these potential microorganisms present in the rhizobiome of cocoa-açai intercropping can be a base for future studies concerning plant interaction, microorganisms’ recruitment, and benefits related to the co-cultivation of these plants in the field. 

As the AFS mimics a forest ecosystem, the recruitment of microorganisms by tree roots in the rhizosphere of intercropping plants may be able to select beneficial microbial communities, making this soil microbiome rich in nutrients and with greater ecological diversity [19,24]. The data generated in the present study can reinforce the AFS consolidation in the Brazilian Amazon, making sustainable soil fertilization in cocoa-açai intercropping systems possible by using PGPR. 

Therefore, in this study, we hypothesized that the cocoa-açai intercropping system is different from the cocoa-MS system concerning the microbial community structure and bacterial populations related to plant-growth promotion. Using cultivation-independent methods (PCR-DGGE approach), we aimed to analyze the structure of the microbial community [based on *rrs* gene and intergenic region (ITS) of fungi] and PGPR populations (based on the *alp* gene and n*if*H gene) present in different agriculture systems [AFS, cocoa-MS and, adjacent forest (AF)] in both Amazon seasons (rainy and dry). In addition, the bacterial strains isolated from the rhizosphere soil of the AFS were also tested for the biocontrol of phytopathogenic fungi and for their potential to promote plant growth using in vitro assays. 

## 2. Materials and Methods

### 2.1. Field Experiment and Soil Sampling

This study was conducted at Konagano farm (2°28′50″ S/48°16′49″ W), located in the northeastern mesoregion of Pará, in the municipality of Tomé-Açu, state of Pará, Brazil (Figure 1). The farm’s soil presents nutrients and physicochemical features as shown in Table 1 [25]. The soil is of the dystrophic yellow latosol type (low-fertility soil) with a texture that ranges from clayey to medium [26]. Samples were taken during the months of August and February (2018 to 2019), dry (t1) and rainy (t2) seasons, respectively. All sampling procedures and the experimental field design are described in Figure 2. Soil from the MS, AFS, and AF areas were sampled from four individual plants. In total, 48 soil samples were collected during the two sampling periods.

### 2.2. TC-DNA Extraction and PCR Amplification

The TC-DNA was extracted from rhizospheric and bulk soil samples (0.5 g of each sample) from the four replicates of plants from the MS, AFS, and AF at both time seasons using the commercial kit “Power Soil DNA Isolation” (Mobio, Carlsbad, CA, USA). The amplification reactions of the *rrs* gene of the total bacterial community were performed using U968F-GC [27] and L1401R primers [28]. The partial fragments encoding the bacterial alkaline phosphatase enzyme (*alp* gene) were amplified using ALPS-F730 and ALPS-R1101-GC primers [29]. 

For *nif*H amplification (the gene that encodes the bacterial nitrogenase enzyme) a nested PCR was performed using FGPH19 and PolR primers for the first reaction and AQER and PolF-GC primers for the second [30,31]. Fungal amplicons based on the ITS region were also obtained using a nested protocol. The first reaction was carried out with EF4 and ITS4 primers [32,33], and the second amplification using the ITS1f-GC and ITS2 primers [27,33,34]. 

### 2.3. DGGE and Statistical Analyses

Denaturing gradient gel electrophoresis (DGGE) was performed using the “INGENYphorU” system (INGENY, Leiden, The Netherlands). PCR products (15 μL) were applied to DGGEs containing a linear denaturing gradient of urea and formamide varying from 46.5–65% for the *rrs*, *alp*, and *nif*H genes and a concentration of 23–58% for fragments of the ITS gene [35]. The generated matrices were used in the construction of dendrograms based on Pearson’s similarity coefficient by the UPGMA method (unweighted pair group method with arithmetic mean) through the software BioNumerics 7.5. To evaluate the distribution of the samples, principal component analysis (PCA) was performed using the statistical software Past 3.26 and PERMANOVA test (*p* < 0.05) [36].

### 2.4. Isolation of Bacterial Strains

The bacterial strains were isolated from AFS rhizosphere soil samples collected in August (dry season). The soil samples were diluted in phosphate buffer saline (PBS), and appropriate dilutions were subsequently plated in triplicate onto tryptic soy agar (TSA) culture medium supplemented with 0.2% nystatin (0.1 mg·mL^−1^) to avoid fungal growth [37]. Colonies were selected from 10^−4^, 10^−5^ and 10^−6^ dilutions and the number of CFU·g^−1^ of soil was determined to estimate the bacterial population density. Statistical analysis for the CFU counts was performed using Tukey’s test (*p* < 0.05). Cell morphology was determined using Gram staining. Bacterial cultures were stored in triplicate in tryptic soy broth (TSB) with 20% glycerol at −20 °C.

### 2.5. Molecular Identification of Bacterial Strains

Genomic DNA from bacterial strains was extracted following the phenol-chloroform protocol [38]. The genomic DNA extracted from the isolates was compared using BOX-PCR fingerprint to select representative strains for future identification based on sequences of the *rrs* gene using the BOXA1R primer [39]. Cluster analysis of fingerprints was performed using the unweighted pair method with arithmetic mean (UPGMA) based on the similarity of the coefficient data using the BioNumerics 7.5 software package (Applied Maths, Ghent, Belgium). The isolated strains that obtained at least 70% similarity were considered to belong to the same BOX group. The selected isolates were submitted to amplification of the *rrs* gene using the universal primers 8F [40] and 1492R [41] and sequencing was performed in the Applied Biosystems 3500 Series Genetic Analyzer (Applied Biosystems^®^, Waltham, MA, USA). 

### 2.6. Production of Siderophores, Indole Acetic Acid (IAA), and Antimicrobial Substances (AMS); Mineralization of Organic Phosphate; and Solubilization of Inorganic Phosphate

The production of siderophores was tested and a positive reaction was indicated by the formation of a yellow halo around the bacterial colony [42]. The qualitative test to produce IAA was carried out by observing the color change to reddish tones when adding Salkowski’s reagent to the culture’s supernatant [43]. To detect the production of AMS against *Curvularia*, *Colletotrichum*, *Fusarium*, *Pestalotiopsis*, *Pythium,* and *Rizoctonia* fungi, the direct plate pairing test was used [44,45]. The tests to verify the mineralization capacity of organic phosphate were carried out using the calcium phytate culture medium [46]. The inorganic phosphate solubilization test was performed using the culture medium NBRIP (National Botanical Research Institute’s phosphate) added with 0.025 g·L^−1^ of bromophenol blue (BPB) [47]. The presence of a transparent zone around the colonies indicated a positive result for the mineralization and phosphate solubilization tests.

## 3. Results

### 3.1. Analysis of the Microbial Community Structure and Bacterial Guilds Associated with PGPR

The effects of monoculture systems (MS), cocoa-açai agroforestry systems (AFS), and adjacent forest (AF), along with climatic variations [dry season (t1) and rainy season (t2), August and February months, respectively] on the microbial community structure were analyzed using DGGE based on the amplification of *rrs*, *alp*, *nif*H genes, and the ITS region of fungi. The experimental design applied in this study allowed for the investigation of the influence of sampling time (t1 and t2, dry and rainy seasons, respectively) and different systems. The overall microbial community appeared complex, with a large number of bands in the generated profiles regardless of the studied genes (Appendix A). Through band profile analysis and the statistical test PERMANOVA (*p* < 0.05), it was observed that there was no significant rhizospheric effect in the MS and AFS systems within each studied sampling time (t1 or t2). Consequently, all soil samples collected from the MS and AFS (including rhizospheric and bulk soil) were considered samples of the respective studied system for subsequent analyses, totaling eight sample replicates per system. 

Cluster analysis from DGGE profiles revealed that almost all profiles were separated according to sampling time (similarity around 40%), except for the diazotrophic community band profiles, which were separated with a 90% similarity (Appendix A). Based on the PERMANOVA statistical test (*p* < 0.05) (Table 2), time significantly influenced both tested systems. This suggests that the effect of sampling time on the microbial community structure was stronger than that of the systems. When comparing the systems, total bacterial community analysis (Figure 3A.1,A.2) from DGGE profiles using PCA revealed that the MS was closer to an AF compared to an AFS, especially in t1. Similarly, there was greater overlap between AFS and AF groups in t1 compared to t2, and separation of the MS from AFS and AF systems, indicating a change in the bacterial community structure associated with phosphate solubilization (Figure 3B.1,B.2). In the diazotrophic community analysis (Figure 3C.1,C.2), MS, AFS, and AF systems were grouped in t2, suggesting minimal change in the structure of this community among the systems. Regarding the total fungal community (Figure 3D.1,D.2), there was a clear separation between groups for each treatment in both sampling times. A PERMANOVA test (*p* < 0.05) (Table 2) also supported the effects of the system model at both sampling times on the bacterial community structure, specific bacterial groups, and fungal community, as revealed by DGGE profiles and PCA (Figure 3 and Appendix A). Statistically significant differences were observed in these communities concerning the employed system model in t1 and t2. Thus, these results suggest that the system model might directly influence the structure of these communities. Additionally, PCA was able to distinguish the MS system from AFS and AF areas regardless of sampling time. 

In terms of the soil physicochemical features, a variation in the components based on sampling time and the system model (Table 1) with higher concentrations of carbon (C), phosphor (P), potassium (K), and organic matter (OM) in the MS was observed when compared to the AFS and AF. The physicochemical analysis of the soil also showed similar quantities of nitrogen (N) in all three systems, with minimal variation in the AFS and AF compared to the MS (Table 1).

### 3.2. Isolation, Identification, and Characterization of Bacterial Strains

The bacterial strains from the rhizosphere of the cocoa-açai intercropping were cultured on TSA plates, and 48 colonies were selected, characterized, and named according to the following nomenclature: CP, followed by the plant number (1, 2, 3, and 4) and, subsequently, the isolated strain number. The 48 isolated strains from the cocoa rhizosphere were divided into 36 groups (70% similarity) according to the BOX PCR dendrogram (Figure 4). Some groups were formed by several strains, while others by only a single strain. A representative strain from each group was selected for molecular identification based on the *rrs* gene fragment (approximately 900 bp). The isolated strains belonged to 10 different genera: the genera *Agromyces*, *Lysinibacillus*, *Microbacterium*, *Pantoea*, *Roseomonas*, *Staphylococcus*, *Bacillus*, *Leifsonia*, *Paenibacillus*, and *Rhodococcus* (Table 3). About 25% of the isolates were identified as members of the *Bacillus* genus belonging to six different species. Interestingly, 23% belonged to the *Staphylococcus* genus, and 19% belonged to the *Rhodococcus* genus, with some strains identified only at the genus level. The genera *Agromyces*, *Lysinibacillus*, *Microbacterium*, *Roseomonas*, and *Paenibacillus* were the least abundant among rhizobacteria, representing less than 5% of the isolates. Out of the 48 isolated strains, approximately 60% exhibited at least one of the tested growth-promoting characteristics. Of this total, 25% were capable of producing siderophores, 29% could mineralize organic phosphate, 8% could solubilize inorganic phosphate, and 4% were able to produce indole acetic acid (IAA). Among these isolates, 17% produced antimicrobial substances (AMS) against the fungi *Curvularia*, *Fusarium*, *Pestalotiopsis*, *Pythium*, and *Rhizoctonia*. Of all tested strains, two belonged to the genera *Rhodococcus* and *Pantoea* and showed the potential to produce siderophores, mineralize organic phosphate, and solubilize inorganic phosphate. Six out of the eight strains capable of producing AMS (Appendix A), affiliated with the *Bacillus* genus, were retested in quantitative AMS production assays against phytopathogenic fungi. On average, all tested strains showed inhibition percentages ranging from 50% to 100% against all tested fungi (Appendix A). Statistically significant differences (*p* < 0.05) were observed among AMS-producing strains when tested against *Curvularia*, *Fusarium*, *Pythium*, and *Rhizoctonia* fungi using an ANOVA and Tukey’s test. The CP3-40 strain of *Bacillus* stood out among all tested strains, exhibiting 100% inhibition against the tested fungi, except when tested against *Pythium*, where it showed an inhibition percentage of around 70% (Appendix A).

## 4. Discussion

Agroforestry has been demonstrated to be environmentally more sustainable than agricultural monocultures, influencing crop productivity and yield in the field while enhancing nutrient availability by improving soil biota, as seen in agroforestry systems with cocoa [20,48,49,50,51,52]. The selection of shade tree species directly impacts various aspects of nutrient cycling in cocoa agroforestry systems [9,53]. Additionally, microbial communities in this system can substantially vary in space and time, depending on soil type, associated plant species, and management practices [52,54,55,56]. As far as we know, this is the first polyphasic study of microbial communities in the rhizosphere of cocoa-açai intercropping systems in agroecosystems in the Brazilian Amazon.

The fingerprint data, soil physicochemical analysis, and statistical analyses obtained in this study revealed that the sampling time, dry seasons and rainy seasons, respectively, influenced the structure of microbial communities present in the rhizosphere of the MS and AFS. Furthermore, the community structure was influenced by the system model employed. These data align with studies indicating that climatic seasonality is an important factor in changing the structure of microbial communities in agroforestry systems and that these impacts may vary among different microbial groups and associated species [54,57,58]. According to the presented hypothesis, significant variation was observed in the structure of total bacterial communities as well as in communities associated with phosphate solubilization and fungal communities between the MS and AFS in both dry and rainy seasons. Additionally, a stochastic effect was observed in the composition of these systems, possibly due to extreme climatic fluctuations, changes in land use, and chemical characteristics of the soil [59]. Similar results were observed in experiments conducted in cocoa agroecosystems in Peru, where significant effects of a cover crop presence, soil chemical features, and the cocoa genotype on the structure of bacterial and fungal communities were identified and compared to a monoculture [60]. It has also been reported that intercropping models promote greater bacterial abundance and, to some extent, greater species richness when compared to a monoculture, linking this model system to nutrient-rich soils. This indicates that different root exudates produced by associated species may provide important substrates for microbial growth [12,61]. Furthermore, the duration of agroforestry management was also an important factor in analyzing the composition of this community. It has been proven that cocoa agroforestry systems have a significant impact on soil fungal communities, presenting a higher abundance and richness of fungal communities when compared to a monoculture. These communities can also vary considerably with the implementation of different agroforestry models [55,62,63,64,65].

In relation to the diazotrophic community, the structure of this community was significantly influenced in the MS and AFS during the dry season only (t1). The cocoa monoculture (MS) also appeared closer to the AF regarding the structure of this community. These results suggest that the AFS may have a positive effect on the abundance of nitrogen-fixing microorganisms and the availability of metabolically available nitrogen forms for both crops [66,67,68]. In this sense, the complexity of the bacterial community in the soil of agroforestry systems in the Amazon demonstrates being mostly influenced by soil nutrient availability, while the fungal community is more closely linked to variables related to above-ground plant biomass [69]. Thus, it is essential to emphasize that factors such as biogeography, climate, soil nutritional conditions, and adopted management practices, such as the use of pesticides or the degree of shading in cocoa agroforests, play a determining role in the structure of these communities that can be distinctly influenced.

Concerning soil physicochemical features, higher levels of C, P, K, and OM were observed in the MS compared to the AFS and AF, with a reduction in these components due to climatic seasonality. Additionally, it was noticed that soils in the MS were more acidic compared to the AFS and AF. Therefore, these data suggest that the AFS model employed in the Brazilian Amazon tended to show a higher rate of nutrient depletion compared to the MS, which may be a direct result of the species involved in the consorted model, where these crops compete for resources and may lead to a higher demand for nutrients [61,70], posing a long-term challenge for the sustainability of this system. This variation in soil nutritional composition can also be explained by OM decomposition and nutrient release into the soil, as well as increased leaching and surface runoff during the rainy season, especially in the MS [71,72,73,74,75]. Moreover, in this cultivation model, the absence of vegetative cover can impair soil protection and demand a higher application of chemical fertilizers, resulting in nutrient accumulation in the soil due to concentrated nutrient demand in monocultures [72,73,76,77,78].

Regarding N content, a decrease was observed in the MS and AF compared to the AFS. This reduction was even more pronounced due to seasonal variations in climate. This suggests that the AFS system may exhibit greater resistance to climate variations regarding soil N availability. Some studies have highlighted that in agroforestry systems, the shade provided by trees is one of the factors responsible for increasing soil N availability, as the N cycle is less driven by fertilizer application in this system and more by decomposition and mineralization of the plant material stimulated via shading [66,67,79]. Moreover, the loss of N during the rainy season might be lower in consorted crops compared to a monoculture [80,81]. It has also been observed that soil OM and total N content were reduced in the agroforestry system, and the system’s age was also an important factor in this reduction [82]. These data suggest that agroforestry systems appear to be less dependent on fertilizer application, with decomposition and mineralization of the plant material playing a crucial role in nutrient availability such as nitrogen for plants, indicating greater autonomy of these systems concerning soil fertility, although there are still challenges such as specific nutrient depletion.

Previous studies have shown that some agroforestry system models, such as cocoa systems, may share similarities with natural forests, suggesting a possible relationship between the richness and diversity of the microbial community and soil nutrient dynamics [66,82,83,84,85]. However, it has been evidenced that an increase in the diversity of associated species is not the primary determining factor for microbial group abundance and soil fertility variations [67,69,86,87]. Therefore, these data suggest that agroforestry systems, although sharing similarities in terms of vegetation and biodiversity with forests, may differ concerning the abundance of microbial communities and soil nutrient dynamics, depending on the geographic region, climate, associated species, and agroforestry management age. In this regard, the implementation of sustainable management practices that promote microbial diversity and soil health becomes indispensable, aiming to ensure the long-term productivity and sustainability of cocoa agroforestry systems.

The role of interactions between plants and microorganisms plays a fundamental role in agriculture. Various factors, such as abiotic stress, can impair the growth and productivity of various agricultural crops. Thus, the need for environmentally viable methods to reduce these stresses has led to the use of plant growth-promoting rhizobacteria (PGPR) as a sustainable strategy to optimize plant growth and development, enhance agricultural production, and explore the potential for biofertilizer production through the combined use of PGPR [88]. In this study, bacterial strains tested for their plant growth promotion potential belonged to the genera *Agromyces*, *Lysinibacillus*, *Microbacterium*, *Pantoea*, *Roseomonas*, *Staphylococcus*, *Bacillus*, *Leifsonia*, *Paenibacillus*, and *Rhodococcus*, with the genera *Bacillus*, *Staphylococcus*, and *Rhodococcus* being the most prevalent. The results obtained from PGPR characterization tests showed that among the twelve strains belonging to the genus *Bacillus*, six were capable of producing siderophores, mineralizing/solubilizing phosphate, and producing antimicrobial substances against all tested phytopathogenic fungi. Among the nine strains belonging to the genus *Rhodococcus*, six were able to produce siderophores and IAA, and to mineralize/solubilize phosphate. Of the eleven strains identified as *Staphylococcus*, six were capable of solubilizing and mineralizing phosphate, in addition to producing IAA. Strains belonging to the genus *Pantoea* were capable of producing siderophores and antimicrobial substances, and mineralizing/solubilizing phosphate. This genus is recognized for its ability to produce metabolites that stimulate root formation and development and has been characterized for its ability to solubilize phosphate, produce auxins (IAA) andsiderophores, and inhibit phytopathogens [89]. Bacteria from the genera *Bacillus* and *Paenibacillus* are common in rhizospheric soils and are generally involved in atmospheric nitrogen fixation, phosphate solubilization, micronutrient absorption, phytohormone production, siderophores, and various antimicrobial substances [90,91,92].

It has been demonstrated that the inoculation of açai plant rhizobacteria in açai seedlings can reduce the sensitivity of the seedlings to water deficit, resulting in an increased photosynthetic performance and the activation of antioxidant enzymes, contributing to the growth of these seedlings in nurseries. Among these identified PGPR, two belonged to the genus *Bacillus* [19]. Another study also revealed that the effects of inoculating endophytic bacteria belonging to the genus *Bacillus*, isolated from cocoa trees, in sunflower plants can enhance the plants’ ability to make osmotic adjustments under water stress, resulting in a higher accumulation of organic solutes compared to non-inoculated plants [93]. Several studies have already confirmed the successful use of rhizobacteria in cocoa plants [24] and in palms such as oil palm [94], coconut [95], and açai seedlings [19,23].

Bacteria of the genus *Rhodococcus* have been identified as auxin (IAA) and siderophore producers, with the potential to stimulate plant root development [96,97]. Bacteria of the genus *Staphylococcus* are well-known pathogens commonly found in humans and associated with a wide variety of opportunistic infections. However, some species have been isolated from plant rhizospheres, such as wheat, potatoes, and strawberries [98]. Several species of *Staphylococcus* have been associated with PGPR traits such as nitrogen fixation, phosphate solubilization, and auxin production. Members of this genus, isolated from grass rhizospheres and cocoa plants, have been described as potential halotolerant PGPR, demonstrating phosphate solubilization capacity, ACC deaminase activity, and IAA production, promoting corn growth through oxidative stress management [99] as well as producing antimicrobial substances against phytopathogenic fungi such as *Rhizoctonia solani* [100]. It has been reported that the species *Staphylococcus warneri*, isolated from plant rhizospheres, has the potential for siderophore production, IAA production, and nitrogen fixation [22,101]. Bacterial genera *Leifsonia*, *Paenibacillus*, *Lysinibacillus*, *Microbacterium*, and *Roseomonas* have been found in plant rhizospheric soil and have shown promising results regarding phosphate solubilization potential, siderophore production, salt tolerance, and the presence of the *nif*H gene, which encodes one of the subunits of the bacterial nitrogenase enzyme [21,99]. Thus, it is possible to highlight the potential of bacterial genera present in cocoa agroforestry system rhizospheres for application as bioinoculants in plants.

## 5. Conclusions

In conclusion, the analysis of DGGE profiles revealed the complex structure of the microbial community present in the rhizosphere of AFS, MS, and AF systems, regardless of the genes studied. The impact of the sampling time (dry and rainy seasons) proved to be a dominant factor influencing the structures of microbial communities, overshadowing the influence of the agricultural systems themselves. The AFS differs from the MS in terms of the structure of microbial communities, which were also influenced by climatic seasonality. Remarkably, variations in the bacterial community, especially those associated with phosphate solubilization, and the fungal community outlined differences between MS, AFS, and AF. This underscores the potential impact of agricultural practices on specific microbial groups, elucidating a possible relationship between system models and microbial community structures. These findings collectively suggest that while seasonal variations significantly shape microbial community structures, system models, especially MS and AFS systems, play an important role in shaping specific microbial groups. It is important to emphasize that further studies are needed to deepen our understanding of the diversity and functions of microbial communities in the AFS system and, thus, to better understand the interactions between cocoa and açai plants in this context. Regarding the chemical composition of the soil, the AFS system showed a greater similarity to the AF than to the MS. However, higher concentrations of the nutrients C, P, K, and OM were observed in the MS, which may be associated with both intensive chemical fertilization in this system and higher rates of organic matter decomposition. Bacteria isolated from the rhizosphere of cocoa plants associated with açai plantss demonstrated the potential to promote plant growth in in vitro tests, suggesting the need for further research on the combined use of these strains and their application as bioinoculants in plants. Thus, these data provide important new insights for the development and implementation of more effective and sustainable strategies for the cultivation of important crops such as cocoa and açai.

## Figures and Tables

**Figure 1 microorganisms-12-00398-f001:**
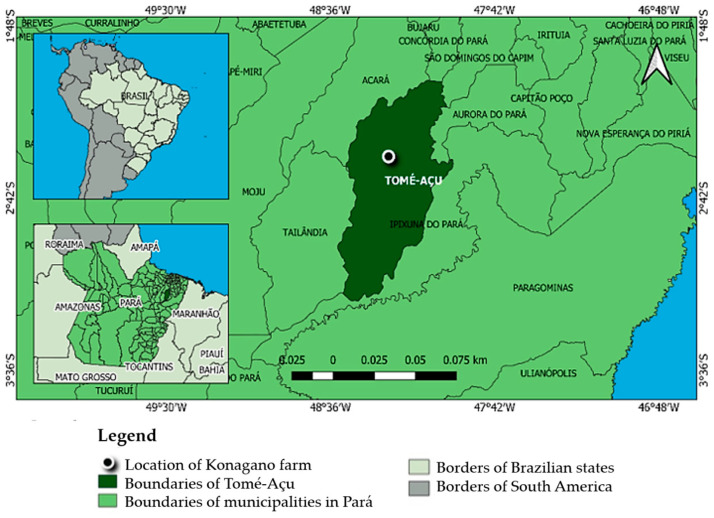
Location map of Konagano farm in Tomé-Açu, Pará, Brazil.

**Figure 2 microorganisms-12-00398-f002:**
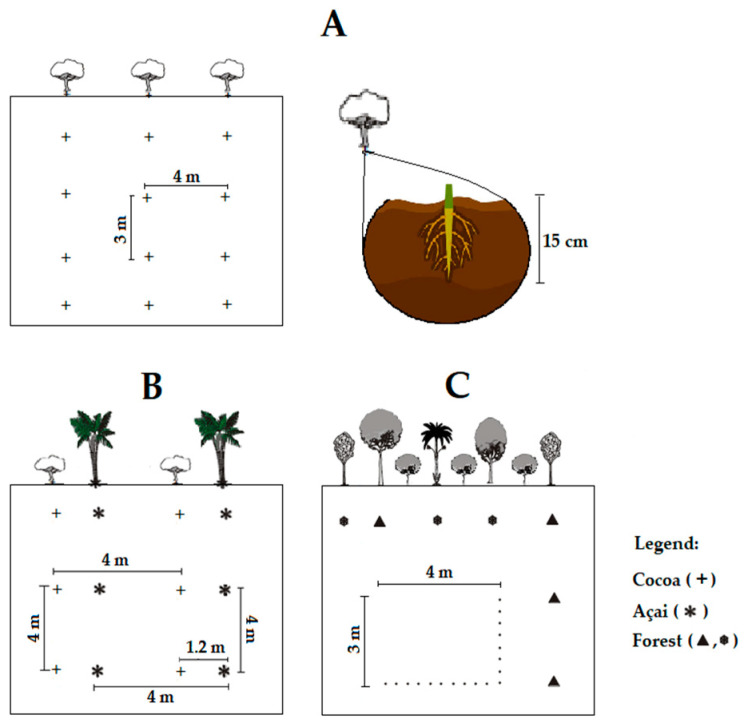
Experimental design of MS (**A**), AFS (**B**), AF (**C**).

**Figure 3 microorganisms-12-00398-f003:**
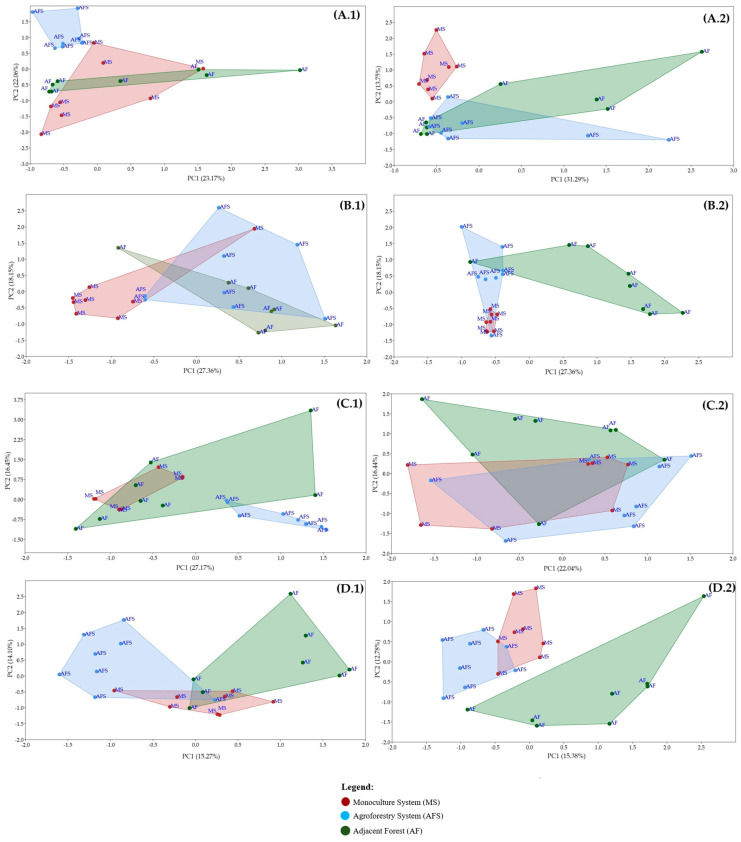
Principal component analysis (PCA) was performed using DGGE patterns of bacterial (**A.1**,**A.2**), phosphate-solubilizing bacterial (**B.1**,**B.2**), diazotrophic (**C.1**,**C.2**), and fungi (**D.1**,**D.2**) genes between cultivation systems (MS, AFS, and AF) at t1 (**1**) and t2 (**2**). The values of the first and second axes indicate the variance (%) explained by each dimension.

**Figure 4 microorganisms-12-00398-f004:**
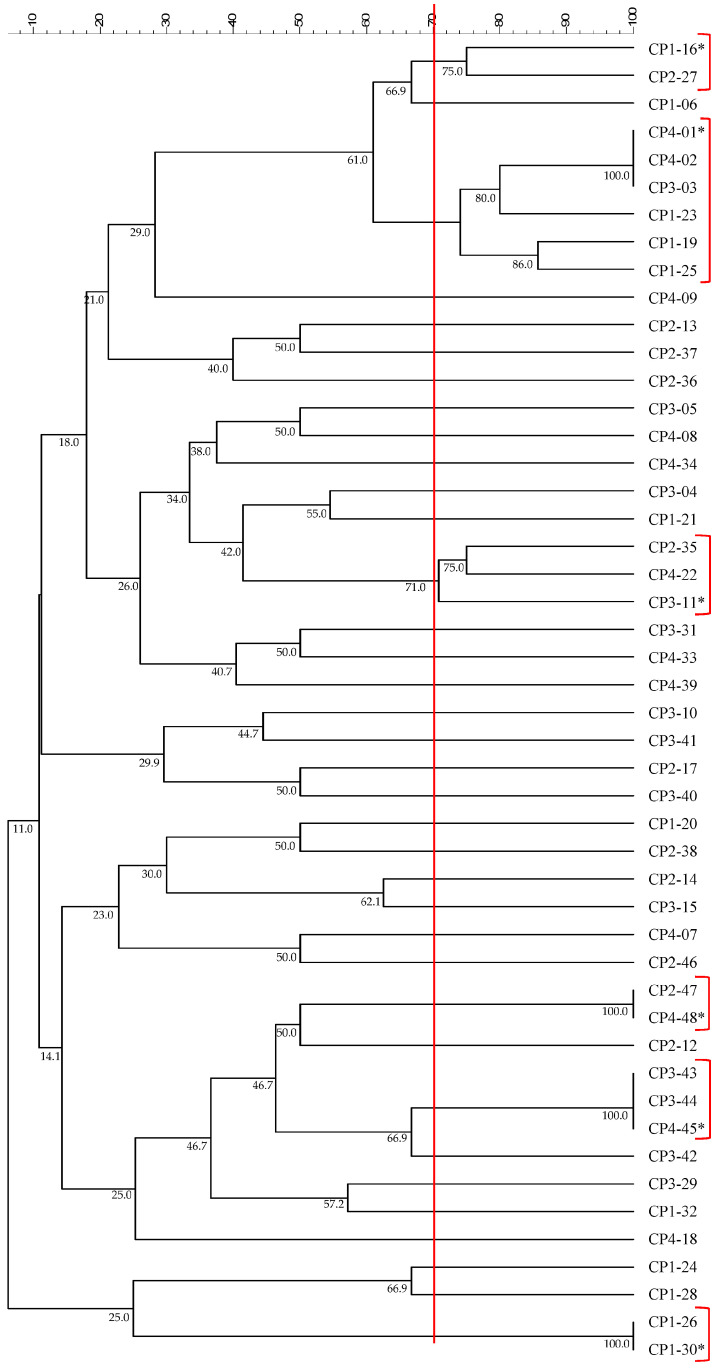
Dendrogram based on the band profile generated by BOX-PCR using the UPGMA method from isolated strains. The 70% similarity value is indicated by the red line and is used to consider a group of strains. Groups with more than one strain are highlighted in red, and the strain representing each group is indicated with an asterisk (*).

**Table 1 microorganisms-12-00398-t001:** Soil physicochemical features by cropping system and sampling time.

Cropping System	Sampling Time	C	MO	N	N	Ratio C/N	P	K	Na	Al	Ca	Ca + Mg	pH
		*g·Kg^−1^			%		**mg·dm^−3^				***cmolc·dm^−3^		H_2_O
Monoculture (MS)	t1	15.2	26.2	0.8	0.1	20.2	86	91	8	0.1	2.3	3.2	5.1
	t2	9.7	16.7	0.3	0.03	32	73	86	3	0.04	2.15	3.07	5.1
Intercropping (AFS)	t1	10.8	18.7	0.9	0.1	12	27	45	11	0.1	4	5.1	6.2
	t2	15.5	15.5	0.8	0.08	11.9	17	33	4	0.07	3.07	4.2	5.09
Forest (AF)	t1	5.2	9	0.6	0.1	8.9	5	28	7	0.2	1.4	1.9	5.7
	t2	7.9	13.7	0.6	0.06	12.3	19	15	6	0.09	2.02	2.84	4.9

*g·kg^−1^: quantity in grams per kilogram of soil; **mg·dm^−3^: quantity in milligrams per cubic decimeter of soil; ***cmolc·dm^−3^: centime of nutrient load per cubic decimeter of soil.

**Table 2 microorganisms-12-00398-t002:** PERMANOVA based on Euclidean distance of DGGE fingerprints, with comparisons between samplings (t1 and t2) or among MS, AFS and AF systems.

Sampling Time	*rrs*	*alp*	*nif*H	ITS
MS × MS (t1 × t2)	0.0008 *	0.0457 *	0.0003 *	0.0001 *
AFS × AFS (t1 × t2)	0.0002 *	0.0007 *	0.0004 *	0.0006 *
**Cropping system**				
MS × AF (t1)	0.0843	0.0033 *	0.1356	0.0006 *
MS × AFS (t1)	0.0015 *	0.0129 *	0.0003 *	0.0015 *
AFS × AF (t1)	0.0006 *	0.9033	0.012 *	0.0006 *
MS × AF (t2)	0.0027 *	0.0015 *	0.1212	0.0009 *
MS × AFS (t2)	0.0012 *	0.0015 *	0.2019	0.003 *
AFS × AF (t2)	0.3702	0.0048 *	0.0063 *	0.0015 *

* Statistically significant when *p* ≤ 0.05.

**Table 3 microorganisms-12-00398-t003:** BOX-PCR groups, identity (%) of representative strains of each BOX group and plant growth-promoting characteristics presented by the 48 bacterial strains isolated from the AFS rhizosphere soil samples.

BOX-PCR Group	Isolate	Closest Database Match (Accession Number), Identity (%)	Siderophore	PO_4_ org	PO_4_ inorg	IAA	AMS				
							*Curvularia*	*Fusarium*	*Pestalotiopsis*	*Pythium*	*Rhizoctonia*
1	CP1-16 *	*Rhodococcus pedocola* (KT301938.1), 99	+	-	-	+	-	-	-	-	-
	CP2-27		+	-	-	-	-	-	-	-	-
2	CP1-06	*Rhodococcus phenolicus* (AY533293.1), 100	+	-	-	-	-	-	-	-	-
3	CP4-01 *	*Rhodococcus sp.* (KT301938.1), 99	+	-	-	-	-	-	-	-	-
	CP4-02		-	-	-	-	-	-	-	-	-
	CP3-03		+	+	+	-	-	-	-	-	-
	CP1-19		-	-	-	-	nd	nd	nd	nd	nd
	CP1-23		-	-	-	-	-	-	-	-	-
	CP1-25		+	-	-	-	-	-	-	-	-
4	CP4-09	*Agromyces indicus* (HM036655.2), 99	-	+	+	-	-	-	-	-	-
5	CP2-13	*Bacillus proteolyticus* (KJ812418.1), 100	-	-	-	-	+	+	+	+	+
6	CP2-37	*Bacillus paranthracis* (KJ812420.1) 100	+	-	-	-	-	-	-	-	-
7	CP2-36	*Staphylococcus hominis* (X66101.1), 100	-	-	-	-	-	-	-	-	-
8	CP3-05	*Lysinibacillus massilienses* (AY677116.1), 100	+	-	-	-	-	-	-	-	-
9	CP4-08	*Pantoea dispersa* (DQ504305.1), 99	+	+	-	-	-	-	-	-	-
10	CP4-34	*Leifsonia aquática* (D45057.1), 99	-	-	-	-	-	-	-	-	-
11	CP3-04	*Roseomonas rosea* (AJ488505.1), 100	+	-	-	-	nd	nd	nd	nd	nd
12	CP1-21	*Staphylococcus epidermidis* (D83363.1), 100	-	-	-	-	-	-	-	-	-
13	CP3-11 *	*Leifsonia poae* (AF116342.1), 99	-	-	-	-	-	-	-	-	-
	CP4-22		-	-	-	-	-	-	-	-	-
	CP2-35		-	-	-	-	-	-	-	-	-
14	CP3-31	*Agromyces indicus* (HM036655.2), 99	-	-	-	-	-	-	-	-	-
15	CP4-33	*Leifsonia aquática* (D45057.1), 99	-	-	-	-	-	-	-	-	-
16	CP4-39	*Staphylococcus homini*s (X66101.1), 98	-	+	-	-	-	-	-	-	-
17	CP3-41	*Pantoea dispersa* (Q504305.1), 98	+	+	+	-	+	-	-	-	-
18	CP2-17	*Staphylococcus warneri* (L37603.1), 99	-	-	-	-	-	-	-	-	-
19	CP3-40	*Bacillus clarus* (KNR180213.1), 99	-	-	-	-	+	+	+	+	+
20	CP1-20	*Paenibacillus terrigena* (AB248087.1), 99	-	-	-	-	-	-	-	-	-
21	CP2-38	*Bacillus paranthracis* (KJ812420.1), 100	-	-	-	-	+	+	+	+	+
22	CP2-14	*Staphylococcus warneri* (L37603.1), 99	-	+	-	+	-	-	-	-	-
23	CP3-15	*Pantoea dispersa* (DQ504305.1), 99	-	-	-	-	-	-	-	-	-
24	CP4-07	*Staphylococccus epidermidis*. (D83363.1), 99	-	+	+	-	-	-	-	-	-
25	CP2-46	*Bacillus aerophilus* (AJ831844.2), 99	-	+	-	-	-	-	-	-	-
26	CP2-47	*Bacillus thuringiensis* (D16281.1), 100	-	-	-	-	+	+	+	+	+
	CP4-48 *		-	+	-	-	-	-	-	-	-
27	CP2-12	*Lysinibacillus xylanilyticus* (FJ477040.1), 100	-	-	-	-	nd	nd	nd	nd	nd
28	CP3-43	*Staphylococcus* sp. (D83363.1), 99	-	+	-	-	-	-	-	-	-
	CP3-44		-	-	-	-	-	-	-	-	-
	CP4-45 *		-	+	-	-	-	-	-	-	-
29	CP3-42	*Paenibacillus glucanolyticus* (AB073189.1), 100	-	-	-	-	-	-	-	-	-
30	CP3-29	*Bacillus aerophilus* (AJ831844.2), 100	+	-	-	-	-	-	-	-	-
31	CP1-32	*Microbacterium testaceum* (X77445.1), 99	-	-	-	-	+	+	+	+	+
32	CP4-18	*Staphylococcus epidermidis* (D83363.1), 99	-	+	-	-	-	-	-	-	-
33	CP1-24	*Bacillus tropicus* (KJ812435.1), 99	-	+	-	-	-	-	-	-	-
34	CP1-28	*Bacillus tropicus* (KJ812435.1), 98	-	+	-	-	+	+	+	+	+
35	CP2-50	*Staphylococcus hominis* (X66101.1), 98	-	-	-	-	-	-	-	-	-
36	CP1-26	*Bacillus wiedmannii* (KU198626.1), 99	-	-	-	-	-	-	-	-	-
	CP1-30 *		-	-	-	-	+	+	+	-	+

PO4 org, organic phosphate mineralization; PO4 inorg, inorganic phosphate solubilization; Siderophore, production of siderophores; IAA, production of indole acetic acid; AMS, production of antimicrobial substances. * Representative strain. nd—Not determined.

## Data Availability

The nucleotide sequences of the bacterial strains determined in this study were deposited in the GenBank database under access numbers MT535606-MT535642, and registered in the Brazilian research database system SisGen (Sistema Nacional de Gestão do Patrimônio Genético e do Conhecimento Tradicional Associado).

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
