# Peer review of "The Microbial Community Structure in the Rhizosphere of Theobroma cacao L. and Euterpe oleracea Mart. Is Influenced by Agriculture System in the Brazilian Amazon"

_microorganisms, 2024, doi:10.3390/microorganisms12020398_

Round 1

Reviewer 1 Report

Comments and Suggestions for Authors

Dear Authors,

Regarding to the manuscript: microorganisms-2802296

    Title:  The microbial community structure in the rhizosphere of cocoa-açai plants is influenced by agriculture system in the Brazilian Amazon

-       -  The main aim of the study is addressed and will be useful for the               researchers who will study in this area.

-       -   The topic is original and relevant to the field.

-        -       The methods are adequate.

-        -   The tables and figures are sufficient.

-        Some specific comments should be taken in consideration to improve the quality of the manuscript:

-       -      The abstract need to be rewritten with more results details.

- -   The sentences should not be started with abbreviation; it is better to start with complete form, please correct all over the manuscript.  

   - There are some writing mistakes, so the manuscript should be         revised carefully

   - It is better to add Figure 4 to the results more than being in the     supplementary materials

-   -  The conclusion need to be rewritten with more results details.

-  - There are several comments provided in the attached manuscript    should be taken in   consideration.

              Best regards

Comments on the Quality of English Language

Minor editing of English language required

Reviewer 2 Report

Comments and Suggestions for Authors

The reviewed work determines the effects of cocoa and acai cacao cultivation on the microbial community structure and bacterial populations associated with plant growth in the rhizosphere of these species. For this purpose, the microbial communities in Amazonian ecosystems during the rainy and dry seasons were analyzed. The paper is typically in forest soil microbiology using standard methods. The work undoubtedly fits the scope of the journal.

The work was well done, and the results were presented. I have a few comments, which I set out below:

1. suggest a title: 

Microbial community structure in the rhizosphere of Theobroma cacao L. and Euterpe oleracea Mart. under the influence of agriculture in the Brazilian Amazon.

2. If you accept the proposed title, remove the plant names from the keywords to avoid repetition.

3. List the months of the dry and wet seasons, as not all readers will be familiar with them.

4. You investigated soil features but not soil composition. So, could you change it to soil features?

5. Give some information about the properties of soil as a medium for microorganisms in an abstract. The name of the soil should be given in the materials section according to WRB. I have not found such information in the main text.  

6. The soil features in Table 3. (Supplementary Materials) must be incorporated in the main text.

7. The soil method analyses?  

8. The conclusion needs to be more precise and not summarise the results. It should be presented in bullet points to improve its comprehensibility and logical structure.

Reviewer 3 Report

Comments and Suggestions for Authors

The article presents intesresting results, however, revision is required for improvements, 

Abstract is short and donot provides enough significant results, moreover, at the end of the abstract, a one liner should be added to highlight the importance of the findings and future prospects.

Keywords---donot use plant scientifc names as keywords

Abstract.....present the hypothesis in a better way and highlight the objectives and at the end of the abstract provide the importance (one liner) of this experiment.

MM....nothing is provided about statistical analysis in MM...should be added in a separate heading...should not be explained in the results

Results....Several PCoA figures created....I think authors are not good enough in statistical applications,,,,a single PCoA figure showing all the components with multivariate analysis was the best way to illustrate all...

Discussion and Conclusion,,,,OK

Comments on the Quality of English Language

OK, see typo errors
